# In Situ Tensile Testing under High-Speed Optical Recording to Determine Hierarchical Damage Kinetics in Polymer Layers of Flax Fibre Elements

**DOI:** 10.3390/polym15132794

**Published:** 2023-06-23

**Authors:** Emmanuelle Richely, Johnny Beaugrand, Michel Coret, Christophe Binetruy, Pierre Ouagne, Alain Bourmaud, Sofiane Guessasma

**Affiliations:** 1INRAE, Research Unit BIA UR1268, 3, Impasse Yvette Cauchois, 44316 Nantes, France; 2Lab Therm & Energie Nantes, LTeN, École Centrale de Nantes, Nantes Université, CNRS, GeM, UMR 6183, 44321 Nantes, France; michel.coret@ec-nantes.fr (M.C.); christophe.binetruy@ec-nantes.fr (C.B.); 3Laboratoire Génie de Production (LGP), Université de Toulouse, INP-ENIT, 65016 Tarbes, France; pierre.ouagne@enit.fr; 4Université de Bretagne Sud, IRDL UMR CNRS 6027, 56100 Lorient, France; alain.bourmaud@univ-ubs.fr

**Keywords:** flax fibre, tensile testing, crack propagation, fractography, high-speed optical imaging, microstructure, in situ tensile testing

## Abstract

This study aims at better understanding the damage and fracture kinetics in flax fibre elements at both the unitary and bundle scales, using an experimental setup allowing optical observation at high recording rate in the course of tensile loading. Defects and issues from flax unitary fibre extraction are quantitated using polarized light microscopy. Tensile loading is conducted according to a particular setup, adapted to fibres of 10 to 20 µm in diameter and 10 mm in length. Optical recording using a high-speed camera is performed during loading up to the failure at acquisition, with speed ranging from 108,000 to 270,000 frames per second. Crack initiation in polymer layers of fibre elements, propagation as well as damage mechanisms are captured. The results show different failure scenarios depending on the fibre element’s nature. In particular, fractured fibres underline either a fully transverse failure propagation or a combination of transverse and longitudinal cracking with different balances. Image recordings with high time resolution of down to 3.7 μs suggest an unstable system and transverse crack speed higher than 4 m/s and a slower propagation for longitudinal crack deviation. Failure propagation monitoring and fracture mechanism studies in individual natural fibre or bundles, using tensile load with optical observation, showed contrasted behaviour and the importance of the structural scale exanimated. This study can help in tailoring the eco-design of flax-based composites, in terms of toughness and mechanical performances, for both replacement of synthetic fibre materials and innovative composites with advanced properties.

## 1. Introduction

Flax is currently viewed as one of the most promising plant fibres to replace synthetic materials, such as glass, in composite industry [1,2]. Thanks to the large specific mechanical performance of flax fibres, light and sustainable technical parts can challenge traditional counterparts [3]. Flax fibres are derived from the stems of the flax plant [4]. It has a hierarchical structure, with a large complexity composed of primary and secondary cell walls. The primary wall is the outermost layer of the fibre [5]. It is composed of cellulose microfibrils, hemicellulose and pectin [6]. The primary cell wall is the origin of the fibre flexibility and its ability to bend and twist [7]. The secondary cell wall is located beneath the primary cell wall, and is composed mainly of cellulose microfibrils and amorphous polymers, such as hemicellulose and lignin [8]. The secondary cell wall is responsible for the fibre’s strength and stiffness, making it resistant to breaking and tearing [9,10]. As is the case for plant fibres, flax usage is threatened by property variability, which is triggered by its genuine phenotype and extraction processes [11]. The effect of this variability on the mechanical performance is amplified by global warming [12]. In fact, the variability of plant fibres within the context of valorisation in the composite industry is threatened by several factors, such as climate change, where the need for quality control is important to guarantee that biobased composite performance meets the specification thresholds. Thus, the understanding of the link between the structure and the performance of plant fibre elements, such as flax, is important to quantify the effect of variability and adopt strategies to attenuate its effect within the context of large-scale manufacturing. Several ways exist to act in that regard, with both leverages at fibre genetics [13,14,15] and extraction parameters. For instance, generation of new porosity and disruption in MFA (microfibril angle) have been reported due to plant fibre extraction machines [16,17]. The damage caused by the mechanical extraction influences the mechanical performance of genuine fibre elements, as well as the fibre-based composites [18,19,20,21]. This damage involves complex deformation mechanisms due to the interaction with the core structure of the fibre, such as the lumen, and the roughness [19,22,23,24]. These mechanisms promote stress localisation phenomena, which are difficult to capture if in situ testing or numerical models that use exact fibre elements ultrastructure are not considered. These defects gain a role at the composite level, through the intrinsic fibre properties or the interfacial properties. Both result in various failure scenarios, depending on the compatibility and the fibre surface treatments used to improve the load transfer between the matrix and the fibre [25]. When it is not possible to use elementary (also called unitary) fibres as reinforcement, bundles potentially close the gap. However, bundles add more complexity, because of the role of the middle lamella [26,27]. Here, at least three main scales need to be investigated to anticipate the composite performance, namely: unitary fibre, bundle and composite microstructure.

Many attempts have been made to understand the unitary fibre and bundle scale on the failure properties of flax [28,29]. By means of scanning electron microscopy and acoustic emission, Romhany et al. [28] showed that flax bundles exhibit a combination of axial splitting of the unitary fibres and radial cracking. Barbulée et al. [30] reported a frictional behaviour of unitary fibres under tension within the flax bundle and the delamination process, which compete with a transverse fibre rupture. Ahmed et al. [31] compared the effect of the unitary fibre and bundle scales on the mechanical behaviour of flax using scanning electron microscopy. A sequential rupture scenario has been suggested, which encompasses bundle rotation, unitary fibres segregation and ultimate failure. Fuentes et al. [24] reported deformation mechanisms of hemp bundles by using digital image correlation. The authors showed that interfacial behaviour explains the ultimate performance of bundles, as a significant shearing is witnessed for weak bundles, and transverse rupture of unitary fibres for strong ones. 

In terms of the role of defects in rupture properties, Mott et al. [32] reported tensile results in wood fibres conducted under SEM observation, where defects acted as preferred sites for failure. Further, Baley [33] demonstrated, through SEM observations, crack initiation from kink bands in flax fibres. Aslan et al. [34] demonstrated the link between variability in properties and the defect distribution in unitary flax fibres. In particular, optical microscopy and SEM results showed a fracturing behaviour originating from both internal and surface defects.

Despite great efforts to quantify the outcome of fracture behaviour in unitary natural fibres and bundles, either by post-mortem or in situ SEM observations, the kinetics and ranking of deformation mechanisms are not easily understood. On top of that, the role of defects in defining rupture scenarios are not thoroughly described. High-speed optical recording has been envisaged as a way to capture the rupture events in natural fibre elements and related composites [25,27,35,36]. These observations highlight the increasing complexity in damage initiation and growth, depending on the scale. Transversely, longitudinal cracking and bridging phenomena are all reported as competing deformation mechanisms, where defects are identified as potential sites for crack initiation. Because of the multiple surface defects, combined with a rapid crack propagation, monitoring failure is challenging because of the difficulty to maintain a high frame rate while observing a large region of interest. Some attempts were made to address this difficulty by performing notches on the fibres prior to testing, to obtain a more localised rupture [35].

With recent progresses in terms of recording velocity by high-speed cameras, it is now possible to decipher the mechanisms involved during the failure in flax fibres without guiding the crack initiation. This study aims at providing a clear view of the deformation mechanisms in flax, at both the unitary fibre and bundle scales, with focuses on the failure scenarios, crack initiation and propagation paths.

This study addresses, firstly, the topography and chemical composition of flax fibre elements, as well as the quantification of the defect density and extent, from polarized light microscopy. Secondly, optical imaging at high acquisition rates is conducted during tensile loading to explore the damage growth in flax fibre elements. The damage mechanisms are discussed according to various cracking models. 

## 2. Experimental Layout

### 2.1. Materials

The Bolchoï variety was selected as a high-fibre yield textile flax fibre for testing. Dew-retted flax stems were provided by Groupe Depestele/Teillage Vandecandelaère. These were cultivated in France and dew-retted in the field in 2017. After harvesting, the plants were stored, scutched and hackled. The fibre bundles obtained were then intentionally damaged by crushing them between rollers on a pilot scutching line (ENI-Tarbes). Therefore, unitary fibres extracted from scutched bundles with contrasted defect density were extracted manually, depending on the location, i.e., close or in-between the marks left by the rollers.

### 2.2. Biochemical Composition

For all biochemical analyses, a first step of homogenization was performed by cryogrinding (SPEX 6700 freezer Mill) approximately 2 g of raw flax bundles into powder, taken from the middle part of the scutched technical flax fibres (about 60 cm long) from each batch. The results given are the mean values, expressed as the percentage of dry matter.

Monosaccharides:

Flax powdered samples (approximately 5 mg per trial) were subjected to pre-hydrolysis in 12 M H_2_SO_4_ (Sigma Aldrich, Darmstadt, Germany) for 2 h at 25 °C. An additional hydrolysis step in 1.5 M H_2_SO_4_ was carried out for 2 h at 100 °C after the addition of inositol, which is used as an internal standard. The alditol acetate derivatives of the neutral sugars [37] were analysed by gas phase chromatography (Perkin Elmer, Shelton, CT, USA), equipped with an DB 225 capillary column (J&W Scientific, Folsorn, CA, USA) at 205 °C, with H_2_ as the carrier gas and a flame ionisation detector. Standard monosaccharide solutions of known concentrations were used for calibration. The galacturonic and glucuronic acids were merged as uronic acids, and determined by the m-hydroxybiphenyl method [38], a colorimetric quantification. Three independent trials were performed for each analysis.

Lignin:

The lignin content was quantified by spectrophotometry following the acetyl-bromide method [39]. Three independent trials were performed for each fibre batch, and the chemicals were laboratory-grade, provided by Sigma Aldrich (Darmstadt, Germany).

Proteins:

The protein content was estimated by the C/N Dumas method, which allows the nitrogen, carbon and sulphur content to be measured by fast and complete combustion of the sample, using an elemental analyser (VarioMicro, Elementar, Lyon, France). The protein content is derived from the nitrogen content using a conversion factor of 6.25, following the general case. The determination of conversion factors is still controversial [40]. Two independent trials were performed.

### 2.3. Sample Preparation

Flax fibre elements, as unitary or in bundles, were glued onto cardboard frames prepared by laser cut, and adapted for an optimal control of the fibre alignment on the tensile bench. Unitary fibres were extracted manually from bundles. All samples used in this study were scanned by optical microscopy prior to tensile testing, in order to characterize their morphology and defect distribution. Optical images were obtained using a microscope (Leitz DMRB, Leica Microsystems, Nanterre, France) equipped with a Hamamatsu digital camera (C11440 ORCA-Flash4.0 LT, Hamamatsu city, Japan). Large image scanning was performed thanks to a motorized stage (Marzhauser, Wetzlar, Germany), allowing scanning along all of the fibres. Images were acquired using both bright light and linearly polarized light, and objectives of ×10 and ×20, leading to a maximum spatial resolution of 18,000 × 2000 pixels, with a pixel size of 328 to 656 nm. The diameters were measured at 3 to 6 locations along each fibre element by optical microscopy. The cross-sectional area was calculated from the mean of the diameters, assuming a circular cross-section. The edges of the paper frame were cut prior to the test.

### 2.4. Optical Image Processing

Optical images were processed to evaluate the density of defects within flax fibres. The process is illustrated in Figure 1. All acquired images were converted from RGB to grey level images. Image superposition is required to encapsulate the grey levels of the polarised image within the fibre element space. For such a purpose, a series of image processing operators, such as threshold, fill hole and max feature isolation, were used to convert the bright image into an outlined one, resulting in the image of fibre boundary extraction as shown in Figure 1. Next, the retrieved boundaries from the outlined image were added and superimposed to the polarised image. The grey level distribution within the fibre, delimited by the added boundaries, were analysed to derive the linear density of the defects and their intensity compared to the mechanical properties.

### 2.5. Standard Tensile Experiments

Unitary fibre scale:

Tensile tests were carried out on unitary fibres up to the fracture point in an MTS machine (MTS System, Créteil, France) at IRDL, Lorient (France), using a 2 N load cell and a strain rate of 1 mm/min, in accordance with the AFNOR NF T 25-501 standard. The controlled testing environment was set to 25 ± 1 °C and 48 ± 2% relative humidity. The fibres were manually extracted and glued to a paper frame with a gauge length of 10 mm. The edges of the paper frame were trimmed prior to testing. The compliance of the system was taken into account, in order to compensate for the influence of non-specimen extension in tensile testing, which leads to underestimation of Young’s modulus and overestimation of strain [41]. Regarding the test reliability, 45 unitary fibres were tested to ensure confidence in measured properties. The stress–strain curves were plotted, taking into account the determination of the cross-sectional areas, which are based on 6 optical measurements along their length and assuming a circular approximation.

Bundle scale:

Tensile testing experiments were carried out on 42 bundles in an MTS machine (MTS System, Créteil, France) at IRDL, Lorient (France), using a 50 N load cell, an acquisition frequency of 10 Hz and a strain rate of 1 mm/min. The controlled testing environment was set to 23 ± 1 °C and 43 ± 1% relative humidity. The fibre bundles were manually extracted and glued to a paper frame with a gauge length of 75 mm. Mean diameters were determined as the average of 6 diameter measurements along the bundles, taken by optical microscopy. The edges of the paper frame were trimmed before testing.

### 2.6. In Situ Tensile Experiments

Tensile testing experiments were carried out on unitary fibres using a Linkam machine TST350 (Linkam Scientific Instruments, Epsom, UK) with a 20 N load cell under high-speed camera recording. The tensile device was placed on a stage equipped with micrometre drives, allowing displacements with a precision in the μm range (Figure 2). As only the central zone could be imaged, the gauge length was adjusted from 15 to 5 mm, in order to capture the damage mechanisms without the need to localize the rupture. Moreover, laser markings were placed at 0.5 mm intervals on the 5 mm cardboard frame, to indicate specific locations of interest noted during the optical scanning. The testing environment was set to 24 ± 1 °C and 27 ± 3% relative humidity. Continuous tensile tests were conducted with a displacement rate of 1 mm/min, in accordance with the AFNOR NF T 25-501 standard. Moreover, the displacement was applied to both jaws. A minimum of 11 unitary fibres and 2 bundles were tested for the damage study.

High-speed camera recording, at speeds ranging from 100,000 to 270,000 frames per second (fps), was achieved using a Photron FASTCAM SA1.1 camera (Photron, Tokyo, Japan), equipped with a telecentric lens (TC16M009, Opto engineering, Mantova, Italy). The full frame dimensions were 1024 × 1024 pixels for frame rates of up to 5400 fps. The dimensions around the fibre elements were reduced to 1024 × 16 pixels, in order to achieve 270,000 fps. The pixel size in this setup is 5.1 μm. The acquisition was triggered manually as soon as the failure was observed, allowing the sequence to be recorded before triggering. The exposure time was reduced to minimize blurring effects. Additional lighting was applied to improve the image quality, using an LED source placed behind the tensile device (Figure 2).

### 2.7. Scanning Electron Microscopy Observations

Scanning electron microscopy (SEM) was performed close to the rupture point on the fibre elements after tensile testing, using a SEM Quattro S (ThermoFisher, Waltham, MA, USA). The fibre elements, still glued on the cardboard frame at both ends, were clamped and observed under low vacuum conditions at a pressure of 100 Pa, an acceleration voltage of 7 kV and a magnification between ×150 and ×15,000.

## 3. Results and Discussion

### 3.1. Biochemical Results

The monosaccharide composition of the Bolchoï variety, determined by gas phase chromatography (GPC), and expressed as a function of dry matter, shows that the main monosaccharide is glucose, which represents 78% of the dry matter. Glucose is often considered to represent the cellulose content of bast fibres, and values between 55 and 90% are reported in the literature for retted textile flax [6,42]. Regarding the other minor in-mass saccharides, the galactose content is 3.6%. Galactose is the main constituent of β(1–4)-galactan, the main encrusting component of unitary fibres, and is likely to be part of rhamnogalacturonan-I (RG-I) [43]. Both have been observed in the secondary cell wall at maturity. The galactose/rhamnose ratio has been reported to adjust the length and number of the RG-I chains, allowing the cellulose microfibril packing to be calculated [44]. In our study, this ratio is close to 4.8. The uronic acid content is 1.9 ± 0.1%, and reflects the presence of pectin in the middle lamella [45].

The lignin content is about 2.4% of the dry matter. These values are in agreement with those reported in the literature, ranging from 1 to 5% for flax [46]. A higher lignin rate has been described by several authors [47,48] as a possible response to an abiotic stress, such as lodging, temperature or lack of water. Since lignin is predominantly found in the middle lamella and cell junctions [49,50], a high degree of retting and a thin middle lamella might explain the relatively low lignin content in the flax studied.

The protein content was estimated from a nitrogen content assay, and shows the same trend as the lignin content, with a low value of around 1.4%. The differences suggest that the compound middle lamella (CML), composed of the middle lamella and adjacent primary cell walls and enriched in structural proteins [43], is less important for the flax studied. However, the origin of the proteins is diverse, and includes both those contained in the fibres (mentioned in the CML) and those introduced by the retting step. It should be noted that retted flax certainly owes its low protein content to the fact that it underwent extensive enzymatic digestion, since proteases are also secreted by the decomposing microorganisms. Moreover, some proteins may be involved in the plant’s defence mechanisms against external stresses [51].

### 3.2. Overall Mechanical Behaviour

Figure 3 shows the typical stress-strain response of the studied flax fibre at both the unitary and bundle scales. At the unitary fibre scale, where the average flax fibre diameter is 16 ± 3 µm, the average Young’s modulus reflecting the compilation of 45 testing results is 40 ± 10 GPa. In the case of plant fibres, due to the natural variability, especially the diameter, presence of kink-bands and size of lumen forms, standard deviations between 20 and 30% are often reported [34]. The observed low stiffness of the flax studied may be related to a high microfibril angle or the presence of a large number of defects. The other properties measured were the elongation at break (2.04 ± 1.05%) and the tensile strength (666 ± 232 MPa). When comparing these results with the literature data [52], the strength and stiffness are positioned in the lower range, while the elongation at break is in the middle range.

Tensile test results at the bundle scale show tensile strengths of 205 ± 85 and 193 ± 92 MPa. Taking into account the mean diameter of the bundles of 114 ± 21 µm, which is approximately 7 times the unitary fibre diameter, the high probability of critical flaws would explain a low strength value at the bundle scale, according to Griffith’s theory [53]. These strength values could be attributed to a high degree of retting, resulting in a less cohesive middle lamella, or to a more technical aspect related to the influence of the gauge length in specimen [15]. Nevertheless, the values obtained are consistent with the literature results for 75 mm gauge length [28,54], and the decrease in strength as a function of the gauge length has already been extensively studied. Indeed, for gauge lengths above a threshold value of about 25 mm, the bundles can be considered as aligned short composites, driven by the weaker mechanical properties of the middle lamella. Load drops prior to the ultimate failure have also been observed as shown in Figure 3, possibly corresponding to the successive early failure of some fibres within the bundles [28,31,55]. The strain at break (1.24 ± 0.34%) appears to be smaller than the unitary fibre scale. This supports the idea of the role of a weaker phase driving the failure of the bundle, namely the middle lamella. The Young’s modulus results (19.1 ± 6.2 GPa) are obtained with a variability of about 32% (i.e., the ratio between the standard deviation and the mean). This variability underlines the change in bundle diameter (18%), but also the non-linear behaviour, against which stiffness assessment is assessed in the last linear segment of the stress-strain curves before the load drops.

The relatively low score for tensile properties at both the unitary and bundle scales may be related to the circular cross-section approximation, which can be better captured by, for instance, automated laser scanning techniques. A correction factor similar to those determined for other types of plant fibres [56,57] could be applied to compensate for the underestimation of the fibre strength and moduli caused by the circular cross-section assumption.

### 3.3. Defect Analysis Results

Figure 4a illustrates the variation in the defect intensity along the fibre length, for which the average size is 19 ± 2 µm. The defect intensity is measured here as the average grey level within the transverse direction. Within the same double Y-axis plot, the variation in the fibre diameter is shown. Large peaks reaching 32 µm indicate the presence of remains of the middle lamella. Smaller peaks of about 12 µm suggest the presence of a twist, which is often encountered during the sample preparation. Attempts to correlate the intensity of the defects with the diameter of the fibres seems to be inconclusive. Further, it appears that the average defect intensity within the entire fibre does not fit the mechanical data, as a large scatter is observed with the tensile strength, for example. However, the measured scatter, which represents the standard deviation of the defect density divided by the average defect intensity, shows a trend with respect to the ultimate properties. The highest scatter appears to be associated with lower tensile stress values.

Figure 4b shows the compiled defect density results for 45 unitary fibres as a function of the average diameter. The diameter is measured along the entire length, as shown in Figure 1, by processing outline images from the bright optical micrographs. The defect density is plotted as the average grey level intensity from optical images acquired under linearly polarized light. Figure 4b shows that there is a large scatter in the results, but a global trend can still be read that supports the idea of increasing defect density with increasing fibre diameter. The scatter also increases as the diameter increases, which can be also related to the difficulty of capturing the defect within the fibre cross-section, unless it is rotated during acquisition. 

Unfortunately, when these data are related to tensile behaviour, there is no general trend between the defect density and the tensile response. Figure 4c shows, however, a negative correlation between the tensile strength and defect density scatter. This scatter is measured as the ratio of the standard deviation to the average defect density along the length of the unitary fibres. 

### 3.4. In Situ Mechanical Behaviour at Unitary Fibre Scale

Detailed characteristics and mechanical properties, obtained from in situ tensile testing of flax unitary fibres and bundles with high-speed camera recordings, are presented in Table 1. Failure sequences were recorded using a high-speed camera, with recording speeds ranging from 108,000 to 270,000 frames per second (fps). The initial, pre-failure and post-failure images were compared with the initial image obtained by bright and polarized light microscopy prior to testing. SEM images of the fracture surfaces after tensile testing are also presented.

The tensile strength and elongation at break of unitary fibres are close to the values obtained from regular tensile experiments for a gauge length of 10 mm, with a mean strength of 656 ± 164 MPa and a mean strain at break of 2.78 ± 1.18%, even though only 11 fibres were tested here. 

The complementary observations lead to different failure scenarios, depending on the fibre, as shown in Figure 5, Figure 6, Figure 7, Figure 8 and Figure 9. In particular, the fracture fibre ends highlight a fully transverse failure propagation in Figure 6, or a combination of transverse and longitudinal propagation along different directions in other cases (Figure 5, Figure 7, Figure 8 and Figure 9). Moreover, the short recording times, down to 3.7 μs, did not allow consecutive images of the failure mechanisms to be obtained in the case of almost complete transverse failure (Figure 5 and Figure 6). This suggests a very unstable system and transverse crack propagation speeds higher than 1.5 and 4 m/s respectively. The longitudinal deviation could therefore lead to a slower overall failure, acting as an energy dissipating mechanism, as hypothesized by Beaugrand et al. [58]. However, observations at the surface only prevent crack propagation in the bulk from being captured. Moreover, it should also be remembered that only five fibres were fully described here.

In Figure 5, we can see that failure has occurred in a defected part of the fibre, which appears as a bright and curved area under polarized light prior to loading. However, the precise location of the failure initiation is difficult to evaluate, and therefore we can only assess the failure initiation in a globally defected area. The fracture surfaces reveal a large transverse failure with a small longitudinal deviation, which could be attributed to a sublayer delamination or the presence of a central lumen [59]. Finally, longitudinal cracks close to the fracture surface can be underlined, with a zoom in on a crack bridging phenomenon.

The fracture surfaces in Figure 6 also show a predominant transverse failure, revealing a central dark area, which could correspond to the lumen. In addition, the possible involvement of surface mechanisms is evidenced in the high-speed camera images, with failure occurring in the vicinity of a surface impurity, indicated by blue arrows. The latter could be remnants of residual middle lamella or cortical tissue. Interestingly, the failure did not occur on the initially observed curved area, which is progressively straightened upon tensile testing.

Transverse cracking, followed by large longitudinal crack propagation, is highlighted in the SEM images of Figure 7. Moreover, it is also noted that the failure occurs in a zone of higher fibre diameter, which appears as a bright area under polarized light. This could be the result of pre-test processing damage, due to the compression of the cell walls between the rollers. Moreover, we can also see the removal of a surface impurity, indicated by a white arrow on the right part of the fibre. Regarding the fractured surfaces, a large longitudinal crack and disorganized cellulose macrofibrils can be seen in the top image.

The combination of transverse and longitudinal mechanisms is highlighted on the SEM images in Figure 8. Failure does not occur in the brightest area, and therefore in the zone of higher defect density along the fibre. The fracture surfaces reveal a significant crack at the surface of the upper part of the fibre, and a cell wall split longitudinally into three parts at the lower part of the fibre.

Finally, the fibre shown in Figure 9 has a failure close to a surface protrusion. At first sight, this could be attributed to a remaining part of adherent tissue, probably from the middle lamella or cortical tissue. However, the fracture surface shows the presence of cellulose macro- and microfibrils close to this protrusion, with a misalignment characteristic of a defect which could have caused the failure [60]. Unfortunately, with the observation of the other part of the fibre, it was not possible to confirm this hypothesis. Furthermore, the very low defect content and straight fibres observed in the initial stage before loading could be the result of a pre-tension applied during the sample preparation, explaining the low strain at break value of 0.8%.

In conclusion, the careful observation of the failure sequences in five different unitary fibres reveals different mechanisms involving pure transverse, and therefore more brittle, failure, or a combination of transverse failure and longitudinal splitting, in agreement with previous results reported in the literature [19,34,35]. Fractographic analysis reveals rough surfaces, typically encountered after tensile solicitation [31], and the presence of cracks close to the fracture surface. Moreover, some fibres have been observed to fail in a zone of high defect density, as shown by polarized light microscopy, but this scenario cannot be generalized to all samples. In agreement with the explanation given by Madsen et al. [19], the failure mechanisms may depend on microstructural parameters, such as the severity and spacing of defects, as well as the size of the lumen. Therefore, it is difficult to explain the differences between the fibres and the correlation with their mechanical properties based on the surface information provided in this study only. The use of X-ray tomography will be of interest in the second part of the study to access bulk information and, especially, defect severity and porosity-related mechanisms. Finally, the possible implication of surface contaminants in failure initiation is highlighted in two cases. As evidenced in Figure 9, the surface flaws may also coincide with zones of cellulose microfibril misalignment. As a perspective, bulk information will help to confirm and understand the underlying mechanisms.

### 3.5. In Situ Mechanical Behaviour at Bundle Scale

The investigation continued at the bundle scale, to better understand the contribution of the middle lamella to the failure mechanisms. Two bundles reflecting contrasted behaviour are shown in Figure 10 and Figure 11. The first bundle shows a hierarchical failure sequence characterized by inter-fibre delamination, followed by the transverse failure of some unitary fibres (Figure 10). This is consistent with the observations of several authors [61], who point to fibre splitting prior to complete failure. Details of the rough interface are observed in the SEM images with the residual compound middle lamella and fibrillar structure, which may correspond to cellulose macrofibrils of the outer layers. In contrast, Figure 11 shows a predominant transverse failure of the unitary fibres without delamination. It occurred in a thicker zone, which could correspond to a defected area. However, polarized light microscopy did not allow the observation of defects in thick bundles, and volume information will help decipher the role of porosity and initial delamination in the different behaviours. Indeed, the second scenario lad to an early failure, characterized by a lower strength and strain at break (around 400 MPa and 1.8%, compared to 794 MPa and 3.5% reported for the first bundle). The value of 800 MPa for the strength is consistent with the values reported by Bos et al. [62] for such a short gauge length. As hypothesized by Fuentes et al. [24], the biochemical composition of the middle lamella could also explain the differences between the failure scenarios, which can be altered by moisture content [61,63,64,65]. However, it must be remembered that the gauge length is inferior to most fibre lengths. Therefore, the behaviours observed here may not reflect the loading of the middle lamella as for the longer gauge length, but rather resemble the behaviour of the unitary fibre scale.

## 4. Conclusions

High-speed camera observations during tensile testing revealed the fastest purely transverse failure of some unitary fibres, and a combination of transverse and longitudinal propagation in others, resulting in a delayed failure. Polarized light microscopy revealed a possible involvement of defects, and bright light observations highlighted the possible role of surface effects. However, the underlying mechanisms may be related to porosity, and need further investigation using more adequate tools, such as X-ray tomography. Indeed, surface observation can hardly reveal the extent of the lumen’s role in driving or contributing to the failure at the unitary fibre scale, where transverse failure predominates.

The major limitation of the method is the accessibility to core deformation within the fibres, where, for instance, stress localisation close to lumen space cannot be captured. Another secondary limitation is lack of spatial and time resolutions to capture the cracking behaviour of unitary fibres along the transverse direction. The main advantage is the assessment of fast cracking involving surface defects, allowing revelation of intricate details under high magnification, and allowing the building of realistic scenarios of deformation sequences for a deeper understanding of crack deviation.

However, contrasted behaviours are observed at the bundle scale, linked to the additional complexity induced by the presence of numerous fibres connected by a middle lamella. Indeed, the high-speed camera showed a hierarchical failure by delamination between the fibres, followed by transverse failure of the unitary fibres in one case, and purely transverse failure of fibres close to a defected area in another case. The bulk effects can be again further explained by X-ray tomography experiments, which will be the subject of future work. The idea is to provide realistic scenarios of the transverse rupture for both compact and airy bundles, considering the role of the middle lamella. 

## Figures and Tables

**Figure 1 polymers-15-02794-f001:**
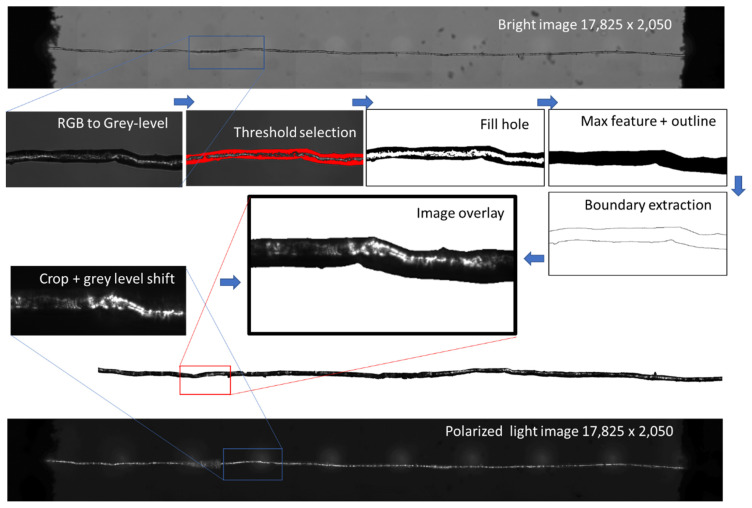
Image processing used to measure the density of defects from optical sources combining bright and linearly polarized lights.

**Figure 2 polymers-15-02794-f002:**
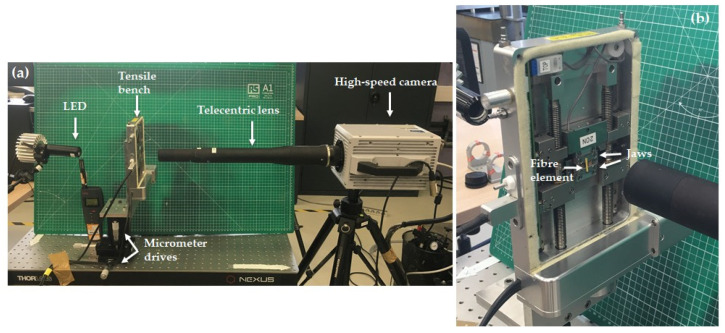
(**a**) Overview of the experimental set up of high-speed imaging in situ tensile testing experiments. (**b**) Magnified view of the specimen mount on the tensile machine.

**Figure 3 polymers-15-02794-f003:**
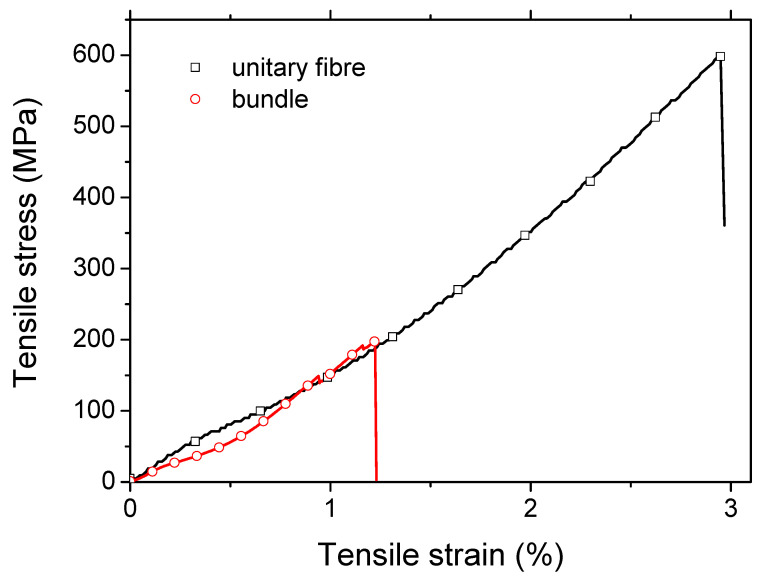
Typical tensile response of flax fibre elements at the unitary and bundle scale.

**Figure 4 polymers-15-02794-f004:**
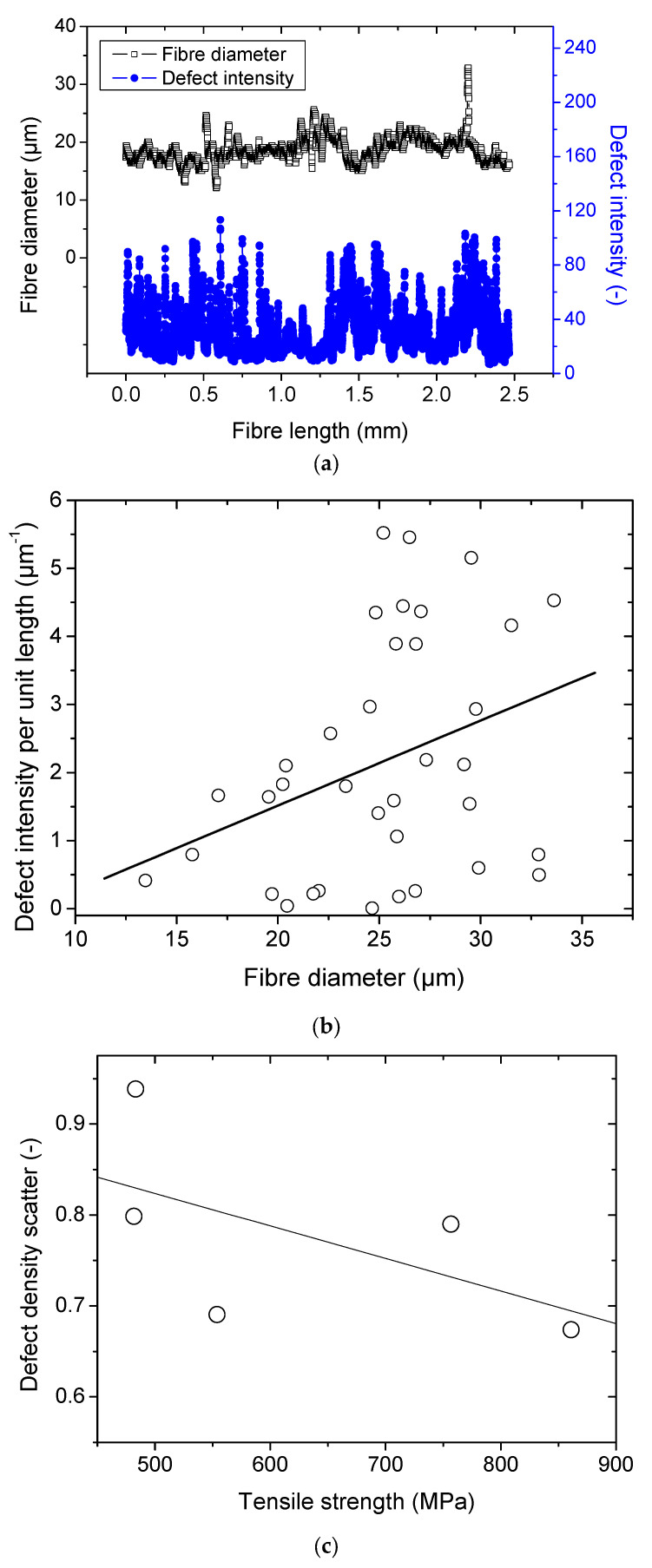
Defect analysis from optical imaging: (**a**) linear density of the defects measured according to the average grey level divided by the diameter, (**b**) correlation between the defect density and the fibre diameter, (**c**) relationship between tensile strength and defect density.

**Figure 5 polymers-15-02794-f005:**
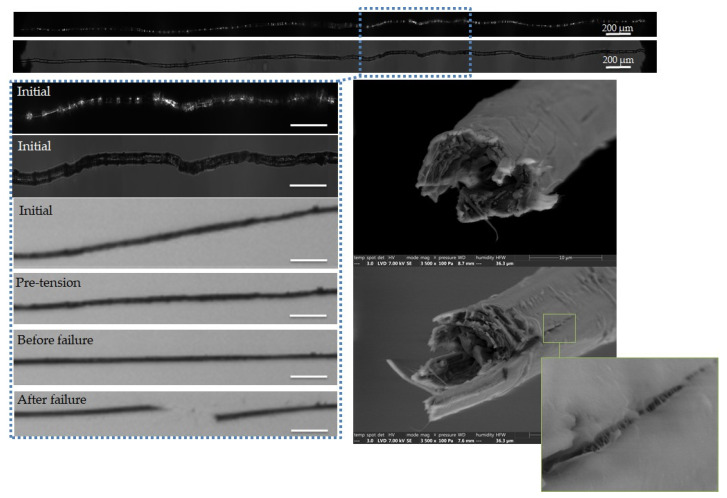
Images of the initial stage observed by optical microscopy under both bright and polarized light, and a zoom on the steps of failure of a flax unitary fibre a-1-11, observed during in situ tensile testing thanks to a high-speed camera. SEM images of the two fracture surfaces after complete failure. The scalebars indicate 100 µm, unless specified.

**Figure 6 polymers-15-02794-f006:**
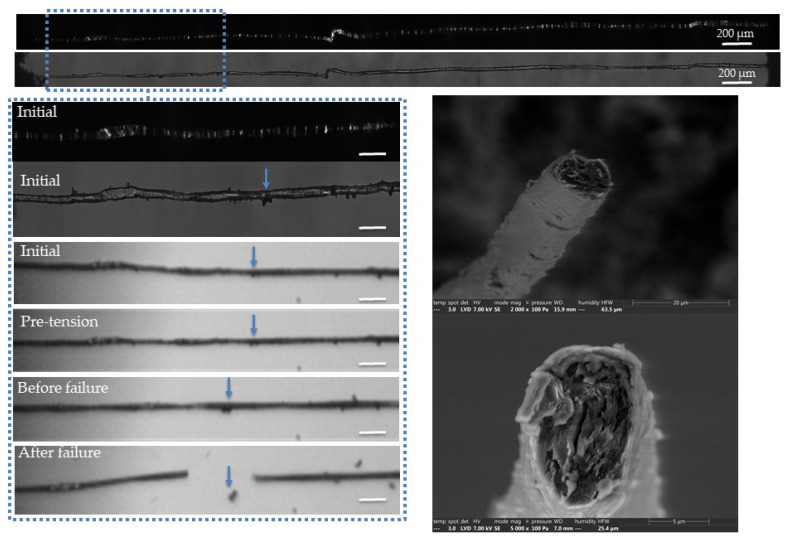
Images of the initial stage observed by optical microscopy under both bright and polarized light, and a zoom on the steps of failure of a flax unitary fibre a-1-22, observed during in situ tensile testing thanks to a high-speed camera. SEM images of the two fracture surfaces after complete failure. The scalebars represent 100 µm, unless specified, and the separation of a surface impurity is indicated by blue arrows.

**Figure 7 polymers-15-02794-f007:**
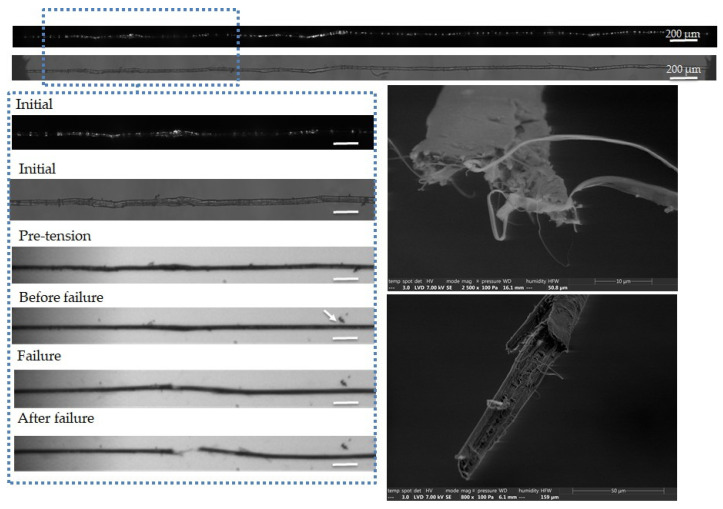
Images of the initial stage observed by optical microscopy under both bright and polarized light, and a zoom on the steps of failure of a flax unitary fibre a-1-14, observed during in situ tensile testing thanks to a high-speed camera. SEM images of the two fracture surfaces after complete failure. The scale bars represent 100 µm, and the separation of a surface impurity is indicated by a white arrow.

**Figure 8 polymers-15-02794-f008:**
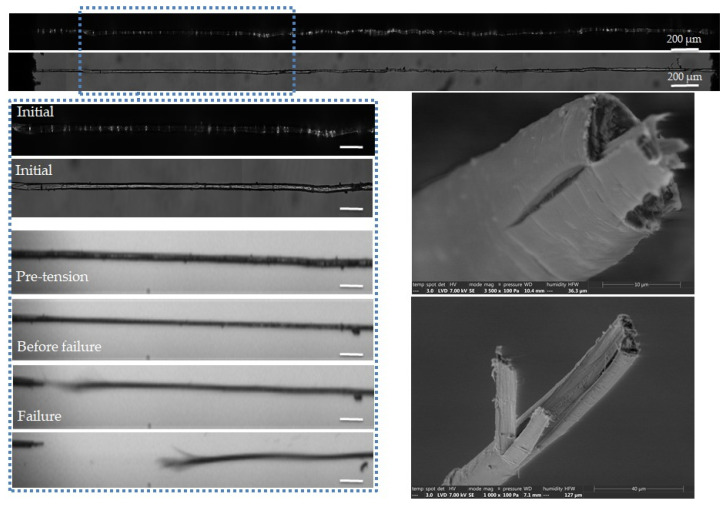
Images of the initial stage observed by optical microscopy under both bright and polarized light, and a zoom on the steps of failure of a flax unitary fibre a-1-18, observed during in situ tensile testing thanks to a high-speed camera. SEM images of the two fracture surfaces after complete failure. The scalebars represent 100 µm.

**Figure 9 polymers-15-02794-f009:**
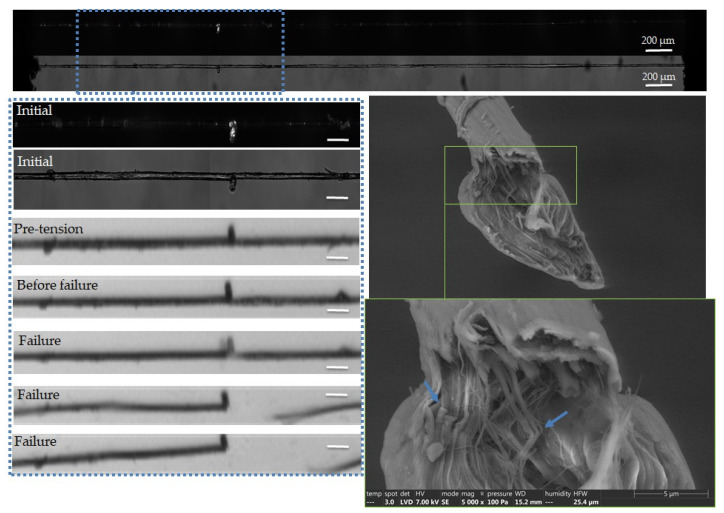
Images of the initial stage observed by optical microscopy under both bright and polarized light, and a zoom on the steps of failure of a flax unitary fibre a-1-20, observed during in situ tensile testing thanks to a high-speed camera. SEM images of the fracture surface of the left part of the fibre after complete failure. The scalebars represent 100 µm, and mesofibril misorientations highlighting a defected area are indicated by blue arrows.

**Figure 10 polymers-15-02794-f010:**
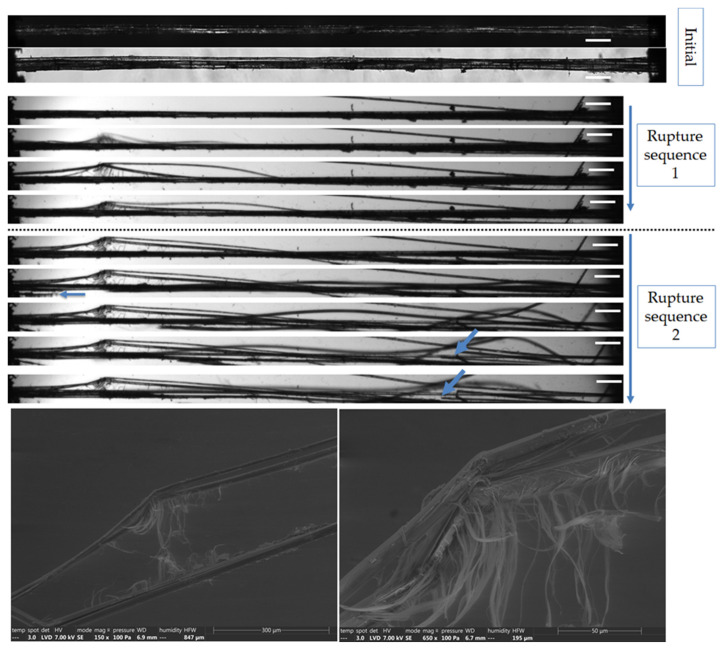
Top: Images of the initial stage observed by optical microscopy under both bright and polarized light, and a zoom on the two consecutive failure sequence of a flax bundle b-1-37, observed during in situ tensile testing thanks to a high-speed camera. Bottom: SEM images taken along the bundle after complete failure. The scale bars represent 200 µm, unless specified, and the blue arrows indicate transverse failures of unitary fibres.

**Figure 11 polymers-15-02794-f011:**
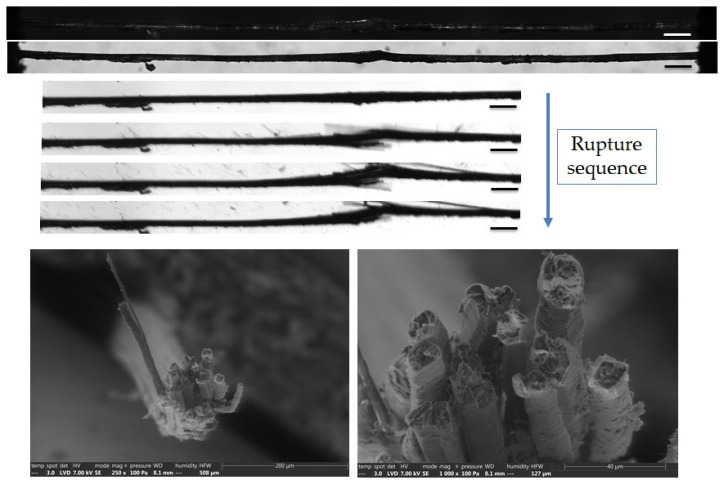
Top: Images of the initial stage observed by optical microscopy under both bright and polarized light, and a zoom on the failure sequence of a flax bundle b-1-41, observed during in situ tensile testing thanks to a high-speed camera. Bottom: SEM fracture surface of one extremity after complete failure. The scalebars represent 200 µm, unless specified.

**Table 1 polymers-15-02794-t001:** Characteristics and mechanical properties of unitary fibres (a) and bundles (b) after high-speed camera in situ tensile testing. ROI: Region Of Interest.

Sample	Recording Speed (fps)	ROI Size (pix.)	Strength (MPa)	Elongation at Break (%)
b-1-41	100,000	704 × 48	403	1.80
a-1-10	108,000	1024 × 48	757	3.19
a-1-11			557	3.76
b-1-37			794	3.45
a-1-12	150,000	1024 × 32	482	3.60
a-1-13			875	1.79
a-1-14			554	2.78
a-1-22			483	3.97
a-1-16			852	1.59
a-1-18			861	3.57
a-1-20	270,000	1024 × 16	462	0.80
a-1-24			595	1.39
a-1-26			741.3	4.17
Mean values for unitary fibres	-	-	656 ± 164	2.78 ± 1.18

## Data Availability

Data Availability Statements are available on demand.

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
