# Peer review of "In Situ Tensile Testing under High-Speed Optical Recording to Determine Hierarchical Damage Kinetics in Polymer Layers of Flax Fibre Elements"

_polymers, 2023, doi:10.3390/polym15132794_

Round 1

Reviewer 1 Report

Dear,

The authors evaluated the natural fiber fracture mechanism using tensile load with optical observation. The manuscript is interesting for fracture mechanics analysis. Therefore, it has merit for publication, with only a few minor tweaks before publication:

> Abstract. Please finish the abstract by showing the importance of the technique and where it can be applied;

> Page 7. Line 262 and 264. I am concerned about the reproducibility of the reported results. The usual thing is that the deviation does not exceed 10% of the value of the mechanical property;

> Please address throughout the discussion the importance of analysis for quality control of natural fibers;

> Please report any limitations of the analysis, as well as advantages;

Moderate editing of English language required

The manuscript needs proofreading in English. There are long sentences that are difficult to read.

Author Response

Dear Reviewer 1

The authors evaluated the natural fiber fracture mechanism using tensile load with optical observation. The manuscript is interesting for fracture mechanics analysis. Therefore, it has merit for publication, with only a few minor tweaks before publication:

> Abstract. Please finish the abstract by showing the importance of the technique and where it can be applied;

Done. Thank you for your kind suggestion that improve the manuscript. We have inserted ‘Failure propagation monitoring and fracture mechanism study in both individual natural fiber or bundles using tensile load with optical observation show contrasted behavior and the importance of the structural scale exanimated. This study can help in tailoring the eco-design of flax based composites in terms of toughness and mechanical performances for both replacement of synthetic fiber materials and innovative composites with advanced properties.’ at the end of the Abstract section. 

> Page 7. Line 262 and 264. I am concerned about the reproducibility of the reported results. The usual thing is that the deviation does not exceed 10% of the value of the mechanical property;

Thank you for this remark; indeed, when synthetic fibres are considered, deviations values for mechanical properties are generally low and under 10% of the value. Things are different in case of plant fibres; due to the natural variability, especially diameters, presence of kink-bands, size of lumen and so on, standard deviations between 20 and 30% are not surprising. This has been addressed by many authors working on the topic, for example:

[1]      Aslan M, Chinga-Carrasco G, Sørensen BF, Madsen B. Strength variability of single flax fibres. J Mater Sci 2011;46:6344–54. doi:10.1007/s10853-011-5581-x.

[2]      Baley C, Gomina M, Breard J, Bourmaud A, Davies P. Variability of mechanical properties of flax fibres for composite reinforcement. A review. Ind Crops Prod 2020;145. doi:10.1016/j.indcrop.2019.111984.

[3]      Lefeuvre A, Bourmaud A, Lebrun L, Morvan C, Baley C. A study of the yearly reproducibility of flax fiber tensile properties. Ind Crops Prod 2013;50:400–7. doi:10.1016/j.indcrop.2013.07.035.

We propose to address this point in the manuscript.  A sentence is proposed.

> Please address throughout the discussion the importance of analysis for quality control of natural fibers;

We believe that we already introduced the context about quality control related to the variability of the plant fibres in the previous version

“Thus, the understanding of the link between the structure and the performance of plant fibres element such as flax is important to quantify the effect of variability and adopt strategies to attenuate its effect within the context of large-scale manufacturing.”

We propose to dd the additional following sentence: The effect of this variability on the mechanical performance is amplified by the global warming [5]. Line 49, In fact, the variability of plant fibers within the context of valorization in composite industry is threatened by several factors such as climate change, where the need for quality control is important to guarantee that biobased composite performance meets the specification thresholds. Thus, the understanding…

> Please report any limitations of the analysis, as well as advantages;

Thank you for your demand.

Done. We propose to add the following sentences in the Conclusion section: line 517 The major limitation of the method is the accessibility to core deformation within the fibers, where for instance stress localization close to lumen space cannot be captured. Another secondary limitation is lack of spatial and time resolutions to capture the cracking behavior of unitary fibers along the transverse direction. The main advantage is the assessment of fast cracking involving surface defects allowing under high magnification to reveal intricate details and allowing building realistic scenarios of deformation sequences for a deeper understanding of crack deviation.

Comments on the Quality of English Language : Moderate editing of English language required

The manuscript needs proofreading in English. There are long sentences that are difficult to read.

Done. We have run a double-checked proofreading, one made by an English native scientist. All the editing modifications can be tracked in the whole manuscript.

Reviewer 2 Report

This study aims to investigate the damage mechanisms triggered by tensile tests in flax fibers elements. The authors examined failure mechanisms involved at both the single fiber and fiber bundle levels using an inline optical recording technique with high-speed cameras in addition to the conventional scanning electron microscopy.

The work is well written and conceived. Results are reported clearly and adequately discussed with the support of high quality figures.

In a scenario of growing interest in the use of natural fibers to replace conventional ones for the development of new sustainable advanced materials, the apparent advancement of knowledge brought about by this study is appreciable.

Overall, the manuscript can be considered suitable for publication as received.

The language is fine.

Author Response

Dear Reviewer 2 :

Comments and Suggestions for Authors

This study aims to investigate the damage mechanisms triggered by tensile tests in flax fibers elements. The authors examined failure mechanisms involved at both the single fiber and fiber bundle levels using an inline optical recording technique with high-speed cameras in addition to the conventional scanning electron microscopy.

The work is well written and conceived. Results are reported clearly and adequately discussed with the support of high quality figures.

In a scenario of growing interest in the use of natural fibers to replace conventional ones for the development of new sustainable advanced materials, the apparent advancement of knowledge brought about by this study is appreciable.

Overall, the manuscript can be considered suitable for publication as received.

Comments on the Quality of English Language : The language is fine.

Dear estimated reviewer, many thanks for your feedback and kindness.